# Alopecia areata is not a risk factor for heart diseases: A 10-year retrospective cohort study

**Heera Lee, You Chan Kim, Jee Woong Choi** *

Department of Dermatology, Ajou University School of Medicine, Suwon, Korea

* dermaboy@gmail.com

## Abstract

Alopecia areata (AA) is an autoimmune skin disease caused by chronic inflammation of hair follicles. Chronic inflammatory skin diseases such as psoriasis and lupus erythematosus can increase the risk of cardiovascular diseases. However, the relationship between AA and heart diseases (HDs) remains unclear. Therefore, we conducted this retrospective cohort study to evaluate the risk of subsequent HDs in patients with AA. We reviewed 3770 cases of AA and from 18,850 age, sex, and income level-matched controls from the National Health Insurance Service-National Sample Cohort. In the subgroup analysis, patients who suffered from alopecia totalis, alopecia universalis, and ophiasis were designated as patients with severe AA and patients having the disease for over a year were designated as patients with long-standing AA. As a result, we found that AA was not associated with a higher risk of heart failure, angina pectoris, or myocardial infarction. There was no significant increase in the risk of overall HD associated with AA (adjusted hazard ratio: 1.17; 95% confidence interval: 0.93–1.48; $p$ = 0.177). Neither the severity nor the duration of AA was related to an increased risk of HDs. During the study period, AA patients did not show a significantly higher cumulative incidence of HDs than controls (log-rank $p$ = 0.157). In conclusion, AA does not increase the risk of HD.

## Introduction

Alopecia areata (AA) is an autoimmune disorder that results in non-scarring hair loss due to an inflammatory response in the hair follicles. Particularly, body attacking its own hair bulbs through CD8+ T lymphocytes is thought to be the primary cause of the disease [1]. Recent studies have revealed that AA is related to autoimmune diseases mediated by helper T cells such as atopic dermatitis, lupus erythematosus, and thyroid diseases [2–7].

Patients with AA show various degrees of involvement including generally well-demarcated patchy loss in AA, loss of all hair on the scalp in alopecia totalis (AT), and complete loss of hair on the scalp as well as on the body in alopecia universalis (AU) [8]. As AA involves a considerable area of the affected body parts, AA patients experience lowered self-esteem and consequently, suffer from various psychiatric problems [9,10]. As AA is related to many other immunologic disorders and is directly connected to patients' quality of life, AA and its comorbidities have garnered nationwide attention.

**Data Availability Statement:** Due to ethical restrictions by the National Health Insurance Service (NHIS) of Korea, the data underlying this study cannot be made publicly available. However, it is available upon request from the Review

Committee of Research Support in NHIS by completing the application form and submit a proposal for review at http://nhiss.nhis.or.kr/bd/ab/bdaba021eng.do.

**Funding:** This study was supported in part by the 2020 Amorepacific Grant. No additional external funding was received for this study.

**Competing interests:** The authors have declared that no competing interests exist.

Increasing evidence suggests that chronic skin inflammatory diseases like atopic dermatitis, psoriasis, and vitiligo are associated with an increased risk of cardiovascular disease [11,12]. In this context, we sought to clarify whether AA patients have a higher risk of cardiac diseases such as ischemic heart disease and heart failure. Some studies claim that AA patients are vulnerable to cardiovascular diseases [13,14]. As AA patients showed increased cardiac marker, we postulated that T cells affecting hair bulbs might extend to other systems. However, Huang et al. [8] suggested that AA patients had decreased risk for stroke and acute myocardial infarction, though the results were not statistically significant. We postulated that such differences could originate from other factors such as racial differences. Sensitivity to the metabolic diseases can vary according to patients' racial and ethnic characteristics because of their cultural and genetic differences [15,16]. It is necessary to evaluate the risk of cardiac disease in Asian patients with AA.

We conducted the present retrospective cohort study using administrative healthcare data. The objective of this study was to determine the risk of heart diseases (HDs) in Korean AA patients.

## Methods

### 1. Data source

The present study was conducted using data from the Korean National Health Insurance Service (NHIS)-National Sample Cohort from 2002 to 2013 [17]. A total of 1,125,691 participants were randomly selected from 97.1% of the Korean population enrolled in the NHIS and were followed up for 12 years. The diagnosis of the cohort was based on the International Classification of Diseases, Tenth Revision (ICD-10). The data consisted of encrypted personal information, diagnostic codes, prescribed drugs, and medical costs. The validity and the usefulness of the cohort data has been verified in several studies [10,18,19].

### 2. Study population and covariates

From the sample cohort, subjects with ICD-10 codes for AA (L63 and all its sub-classification codes including AT [L63.0], AU [L63.1], ophiasis [L63.2], other AA [L63.8], and unspecified AA [L63.9]) were included in the AA cohort. Cohort entry date was defined as the day when the patients received these codes for the first time. To obtain a control group, age, cohort entry date, sex, and income level-matched random samples who had not been diagnosed with AA codes were included (1:5 ratio of patients and controls).

The study subjects were followed up until they were diagnosed with HDs such as heart failure (HF), angina pectoris (AP), acute myocardial infarction (AMI), and chronic myocardial infarction (CMI) with the corresponding ICD-10 codes (HF [I50 and its subclassification codes], AP [I20 and its subclassification codes], AMI [I21, I22, I23, and I24 and its subclassification codes], CMI [I25 and its subclassification codes])during the observation period (from January 2004 to December 2013). Subjects who did not develop HDs were followed up until death, emigration, or the end of December 2013, whichever event occurred first. For accuracy, we selected patients who had these diagnostic codes assigned only by dermatologists in case of AA and by internists in case of HDs. Subjects who had HD codes before receiving any AA code, who had AA codes during the washout period (from 2002 to 2003), or those aged less than 40 years were excluded. Subjects with systemic comorbidities such as hypertension, diabetes, and dyslipidemia before the cohort entry date were identified. Additionally, HD-related comorbidities including stress disorders, peripheral vascular diseases, atherosclerosis, and stroke were also assessed due to their potential as confounders. In subgroup analyses, we further defined severe AA group as patients with AT, AU, or ophiasis diagnostic codes (other AA

patients were categorized into mild AA group). We also designated a long-standing AA group, which was defined as patients with AA for over a year with more than ten outpatient visits (other AA patients were categorized into episodic AA group). A schematized flow chart of the study is shown in Fig 1.

### 3. Statistical analysis

Initially, we used the chi-square test to categorize and to evaluate the statistical significance of the data as a descriptive analysis. After adjusting for significant differences, we obtained hazard ratios (HRs) and 95% confidence intervals (CIs) by Cox proportional hazard regression analysis. Only the results with a two-tailed $p$-value less than 0.05 were considered significant. In addition, the Kaplan-Meier method was used to estimate the risk of HD. Statistical analyses were performed using SAS 9.2 (SAS Institute Inc., Cary, NC, USA) and MedCalc Statistical Software 18.9.1 (MedCalc Software Ltd., Ostend, Belgium).

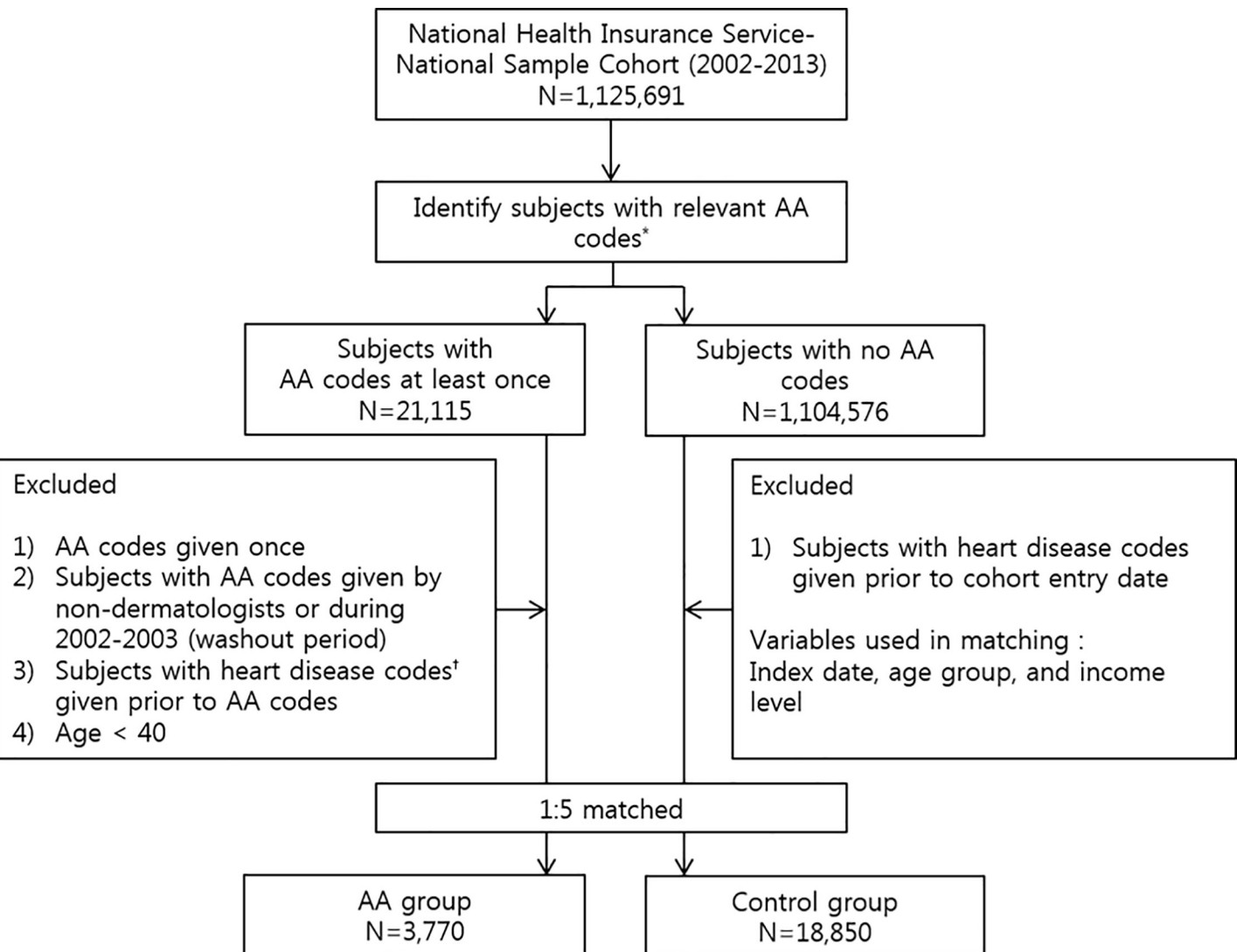

**Fig 1. Study flowchart.** AA, alopecia areata; AT, alopecia totalis; AU, alopecia universalis; HD, heart disease. *AA codes: L63 and its sub-classified codes. †HD codes: I20 (angina), I21-I25 (myocardial infarction), I50 (heart failure).

### 4. Ethical statement

The present study used de-identified data and was approved by the Institutional Review Board of the Ajou University Hospital (IRB-No: AJIRB-MED-EXP-18-404).

## Results

### 1. Demographic and clinical characteristics of the study population

During the observation period, 3770 AA patients and 18,850 matched controls (a ratio of 1:5) were selected from the sample cohort of 1,125,691 participants. The prevalence of dyslipidemia was significantly higher in the AA group when compared with the control group. However, there were no significant differences in other comorbidities between the two groups (Table 1).

### 2. Incidence of HD in AA patients and control subjects

The most common HD comorbidity was AP, followed by CMI, HF, and AMI in both groups. In the univariate analysis, the HRs for HF, AP, and CMI were higher in the AA group when compared with the control subjects. However, the result was not statistically significant. After adjusting for confounders, the HR remained insignificant.

The highest HR was found for HF (HR: 1.75; 95% CI: 0.93–3.29) in the univariate analysis. After adjustment, the highest HR was observed for HF (HR: 1.74; 95% CI: 0.93–3.28). However, both of these results were not statistically significant.

During the follow-up period, the overall incidence of HDs in the control and in the AA groups was 1.99% and 2.36%, respectively. The risk of developing at least one HD was higher

**Table 1. Demographic and clinical characteristics of the study population.**

| Characteristics | AA patients | Control subjects | *P* value |
|---|---|---|---|
| Case No. | 3770 | 18850 | |
| Sex, N (%) | | | 1.000 |
| Men | 1662 (44.1) | 8310 (44.1) | |
| Women | 2108 (55.9) | 10540 (55.9) | |
| Age distribution, N (%) | | | 1.000 |
| 40–59 | 3283 (87.1) | 16415 (87.1) | |
| 60–79 | 478 (12.7) | 2390 (12.7) | |
| 80- | 9 (0.2) | 45 (0.2) | |
| Income level, N (%) | | | 1.000 |
| Lower third | 778 (20.6) | 3890 (20.6) | |
| Middle third | 1187 (31.5) | 5935 (31.5) | |
| Upper third | 1805 (47.9) | 9025 (47.9) | |
| Common systemic comorbidities, N (%) | | | |
| Hypertension | 574 (15.2) | 2959 (15.7) | 0.476 |
| Diabetes | 224 (5.9) | 1138 (6.0) | 0.849 |
| Dyslipidemia | 169 (4.5) | 676 (3.6) | **0.009** |
| Other HD related comorbidites, N (%) | | | |
| Stress disorder | 25 (0.7) | 90 (0.5) | 0.163 |
| Peripheral vascular diseases and atherosclerosis | 163 (0.9) | 40 (1.1) | 0.255 |
| Stroke | 87 (2.3) | 402 (2.1) | 0.503 |

HD, heart disease.

Bolding indicates statistical significance.

**Table 2. Hazard ratio and 95% confidence intervals of heart diseases in patients with alopecia areata and in control subjects.**

| Heart diseases comorbidities | Event/total | Univariate HR (95% CI) | | P-value | Adjusted* HR (95% CI) | | P-value |
|---|---|---|---|---|---|---|---|
| Heart failure | | | | | | | |
| Control subjects | 37/18850 | 1.00 | (reference) | | 1.00 | (reference) | |
| AA patients | 13/3770 | 1.75 | (0.93–3.29) | 0.083 | 1.74 | (0.93–3.28) | 0.084 |
| Angina pectoris | | | | | | | |
| Control subjects | 264/18850 | 1.00 | (reference) | | 1.00 | (reference) | |
| AA patients | 62/3770 | 1.17 | (0.89–1.54) | 0.269 | 1.16 | (0.88–1.53) | 0.300 |
| Acute myocardial infarction | | | | | | | |
| Control subjects | 45/18850 | 1.00 | (reference) | | 1.00 | (reference) | |
| AA patients | 4/3770 | 0.44 | (0.16–1.23) | 0.118 | 0.45 | (0.16–1.24) | 0.121 |
| Chronic myocardial infarction | | | | | | | |
| Control subjects | 79/18850 | 1.00 | (reference) | | 1.00 | (reference) | |
| AA patients | 17/3770 | 1.07 | (0.63–1.81) | 0.797 | 1.07 | (0.63–1.80) | 0.804 |
| †All the heart disease occurrence | | | | | | | |
| Control subjects | 375/18850 | 1.00 | (reference) | | 1.00 | (reference) | |
| AA patients | 89/3770 | 1.18 | (0.94–1.49) | 0.158 | 1.17 | (0.93–1.48) | 0.177 |

AA, alopecia areata; CI, confidence interval.

*Adjusted for the variables of significant difference in frequency by Chi square test.

†All patients who developed at least one heart diseases in the table.

in the AA group, but the difference was not significant (adjusted HR: 1.17; 95% CI: 0.93–1.48) (Table 2).

### 3. Cumulative incidence of HD in patients with AA

The Kaplan-Meier curve was plotted to determine the cumulative incidence of HDs in patients with AA and in control subjects. Patients with AA had an increased risk for HDs, although the difference was not statistically significant (log-rank $p$-value = 0.157) (Fig 2).

### 4. Subgroup analysis according to disease severity and duration

Among the 3770 patients, 106 were patients with severe AA who had AT, AU, or ophiasis and 419 patients had long-standing AA. In the subgroup analysis based on the disease severity, no HD occurrence was observed in patients with severe AA. However, HDs occurred in 2.43% of the patients with mild AA. Based on the disease duration, the occurrence of HD events in long-standing AA patients and in episodic AA patients was 2.15% and 2.39%, respectively. The HRs for the HDs in patients with mild AA and in patients with episodic AA were higher than those in matched controls. However, the difference was not significant (adjusted HR: 1.21; 95% CI: 0.96–1.52 for the mild AA subgroup, adjusted HR: 1.22; 95% CI: 0.96–1.55 for the episodic AA subgroup). HR for HDs in the long-standing AA subgroup was lower than that in the control group, but the difference was not significant (adjusted HR 0.89; 95% CI: 0.46–1.73) (Table 3).

### Discussion

In this retrospective cohort study, we found that AA was not related to an increased risk of HD. This is the first study to include such a large cohort of Asian patients with AA to assess the risk for HDs. Recently, a study involving heterogeneous AA population in the USA found

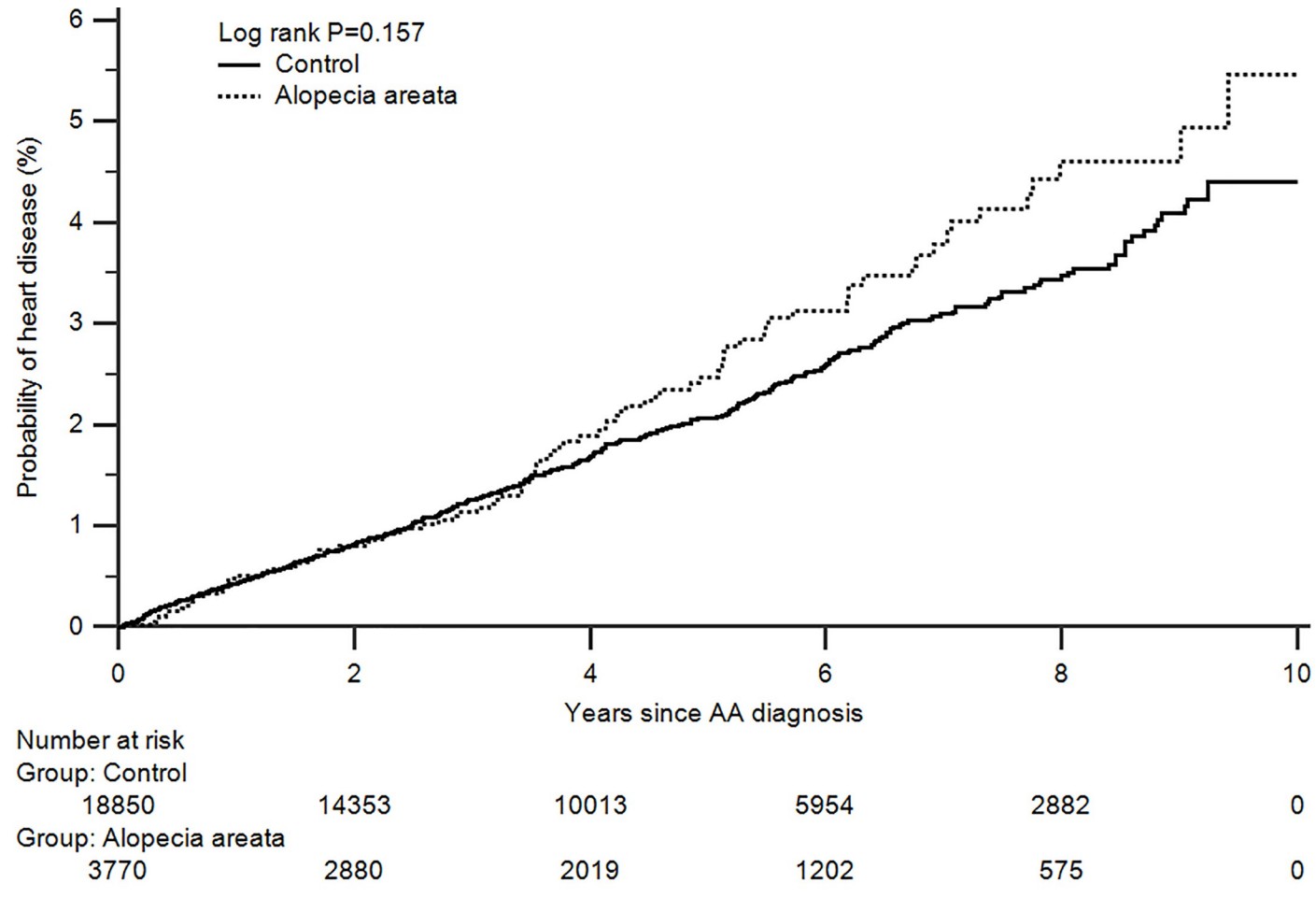

**Fig 2. Cumulative incidence of heart diseases in patients with alopecia areata.** The difference between the two study cohorts was calculated using the log-rank test.

**Table 3. Hazard ratio and 95% confidence intervals of heart diseases according to the subgroup analysis.** Subgroup analysis was performed according to disease severity and duration.

| | Event/total | Univariate HR (95% CI) | | P-value | Adjusted* HR (95% CI) | | P-value |
|---|---|---|---|---|---|---|---|
| By disease severity | | | | | | | |
| Control subjects | 375/18850 | 1.00 | (reference) | | 1.00 | (reference) | |
| Mild AA patients | 89/3664 | 1.27 | (0.97–1.53) | 0.096 | 1.21 | (0.96–1.52) | 0.110 |
| Severe AA patients | 0/106 | NA | NA | NA | NA | NA | NA |
| By disease duration | | | | | | | |
| Control subjects | 375/18850 | 1.00 | (reference) | | 1.00 | (reference) | |
| Episodic AA patients | 80/3342 | 1.23 | (0.96–1.56) | 0.099 | 1.22 | (0.96–1.55) | 0.113 |
| Long-standing AA patients | 9/419 | 0.90 | (0.46–1.74) | 0.746 | 0.89 | (0.46–1.73) | 0.738 |

AA, alopecia areata; CI, confidence interval; NA, not applicable

*Adjusted for the variables of significant difference in frequency by Chi square test.

that AA was not related to an increased risk of AMI. The results were not statistically significant (odds ratio: 0.91; 95% CI: 0.59–1.39) [8]. Though there was a difference in the racial demographic characteristics between the aforementioned study and the present study, there was no discrepancy in the finding that AA patients were not prone to develop HDs. Therefore, it seems that racial differences do not affect the risk of HD.

A causal association between AA and HDs requires the presence of a consistent association in different studies, a dose-response relationship, and biologically plausible mechanisms. However, studies evaluating the risk of HDs in AA patients have not shown such consistency.

C3H/HeJ mice with AA had higher levels of cardiac troponin, a serum HD marker, and showed a higher risk of cardiac hypertrophy [13]. In addition, AA patients had increased HD biomarker cardiac troponin I in a previous report [14]. However, no differences were observed in cardiovascular risk between AA patients and controls in a previous cohort study [8]. Recently, a meta-analysis conducted by Amamoto et al. [20] reported that AA was not related to HDs (relative risk: 0.91; 95% CI: 0.60–1.39; $p = 0.66$). Also, Lee et al. [21] reported similar findings in their meta-analysis. Likewise, there were no consistent findings about the risk of HDs in cohort studies. In addition, the results of all these studies were statistically insignificant. This implies that patients with AA do not have a significantly higher or lower risk of associated HDs.

There was no dose-dependent relationship among AA severity, duration, and incidence of HDs in the present study. Even, there were no cases of HDs in the severe AA group. In addition, the adjusted HR was 0.89 for long-standing AA patients, while it was 1.22 for patients with episodic AA.

Increasing evidence suggests that AA does not extend to systemic inflammation. Psoriasis, which is known to have a significantly higher risk of associated HDs [22], is associated with higher C-reactive protein (CRP) levels. Moreover, serum CRP level is one of the factors indicating the severity of psoriasis [23,24]. However, a recent study reported that levels of serum homocysteine or high-sensitivity CRP were not increased in AA patients [25]. And studies evaluating the risk of HDs in AA patients have not shown consistent results so far. A study by Huang et al. found that AA was not related to an increased risk of acute myocardial infarction without statistical significance [8]. In addition, Lee et al. showed the risk of cardiovascular mortality is not associated with presence of AA [26]. However, Shin et al. claimed significantly increased risk of acute myocardial infarction in patients with AA only in the later stage of the observation [27].

Our results must be interpreted in the context of our study design. Diagnostic data were obtained from a national healthcare database, which could lead to errors. There could be a possibility that enrolled patients had AA or HD before they received diagnostic codes from the database. In order to resolve this problem, we set the first two years as a washout period. Patients who received the AA diagnostic codes during this period were not enrolled in our study. There could be significant differences in demographic characteristics between the AA group and the control group while selecting each member. Therefore, we adjusted for such confounding factors while analyzing the hazard ratios. In conclusion, the results of this large retrospective cohort study suggested that patients with AA did not show an increased risk for HDs.

## Acknowledgments

The present study used the National Health Insurance Service (NHIS) National Sample Cohort data, which were released by the NHIS (NHIS-2017-2-596). The authors alone are responsible for the content and the writing of this manuscript.

## Author Contributions

**Supervision:** Jee Woong Choi.

**Writing – original draft:** Heera Lee.

**Writing – review & editing:** You Chan Kim.

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
