## [Decision Letter · Decision Letter 0]

24 Feb 2021

PONE-D-21-00164

Alopecia areata is not a risk factor for heart diseases: A 10-year retrospective cohort study

PLOS ONE

Dear Dr. Choi,

Thank you for submitting your manuscript to PLOS ONE. After careful consideration, we feel that it has merit but does not fully meet PLOS ONE’s publication criteria as it currently stands. Therefore, we invite you to submit a revised version of the manuscript that addresses the points raised during the review process.

We look forward to receiving your revised manuscript.

Kind regards,

Feroze Kaliyadan, M.D.

Academic Editor

PLOS ONE

Journal Requirements:

3. Thank you for providing the date(s) when patient medical information was initially recorded. Please also include the date(s) on which your research team accessed the databases/records to obtain the retrospective data used in your study.

4. Thank you for stating in the text of your manuscript "The present study used de-identified data and was approved by the Institutional Review Board of the Ajou University Hospital (IRB-No: AJIRB-MED-EXP-18-404)". Please also add this information to your ethics statement in the online submission form.

5. Please provide a sample size and power calculation in the Methods, or discuss the reasons for not performing one before study initiation.

Reviewers' comments:

Reviewer #1: The study examined the data of a large cohort of Asian patients diagnosed with alopecia areata with regard to their risk of developing heart diseases. A few clarifications:

Were smoking/ substance abuse assessed as potential confounders. If potential confounders have been missed, add them as limitations.

The increased number of dyslipidemic patients in the AA group- whether on treatment, potential implications.

The authors may discuss the mechanistics of AA and heart disease, a little more in depth (some authors have hinted at a benefit, as well).

Reviewer #2: In this observational epidemiological study, the authors investigate a possible association between Alopecia Areata (AA) and Heart Disease (HD). They use a prospective cohort with a regression model to understand this relationship. They were able to use a national insurance sample to compare almost 4000 cases with 18850 matched cohorts. No association was found in univariate or multivariate analysis between AA and HD. The paper has the advantages of being a rigorous epidemiological study that does accomplish what it set out to do. The authors were able to find a large cohort and matched samples. Their statistical analysis is credible and their findings are consistent with the available literature in this field. The problem lies in the premise of the study as there is only a tenuous association between AA and HD in the current literature, questioning the need for such an endeavor. The authors also try to understand if there is a relationship between severe AA and HD and there appears to be no basis for considering this. Overall, this study adds to the slim body of evidence that AA does not appear to have any correlation with incidence of HD

Major Issues

Unclear as to how ethnic differences would affect any potential association between AA and HD. The authors briefly mention this but do not elaborate.

Minor Issues

1. Arbitrary definition of long standing AA as greater than one year

2. Authors do not clarify why they chose a 1:5 control ratio and if a smaller (or larger) ratio would have had an impact on the findings

3. Dyslipidemia was more common in AA group but it would have biased the results in favor of an association. Hence, this is likely not relevant

4. ICD code used for heart failure needs to be identified

5. Chronic myocardial infarction is a poorly defined entity. I suspect the authors were referencing ICD I25.2 (old myocardial infarction)

Reviewer #3: This large retrospective cohort study explores the relationship between AA as a chronic inflammatory disorder (like psoriasis, atopic dermatitis, vitiligo…) and heart disease in 3770 patients. Even the subgroups of severe AA (Ophiasis, totalis, universalis) and long standing AA (more than a year per disease duration) failed to show a statistically significant causal association with heart disease (Heart failure, myocardial infarction, angina) and comorbidities over a 9 year study period. I congratulate the author on a well conducted original research paper.

---

## [Author Response · Author response to Decision Letter 0]

26 Mar 2021

Dear Editor,

Thank you for your decision regarding our manuscript.

After we have carefully considered the reviewer's comments, we have revised the manuscript accordingly. We hope that you find this revised version acceptable for publication in your distinguished journal. 

Thank you for your valuable comments.

Reviewer comments:

Reviewer 1 #1: A few clarifications: Were smoking/ substance abuse assessed as potential confounders. If potential confounders have been missed, add them as limitations.

Authors’ response:

- As the reviewer pointed out, smoking could affect the results. Therefore, we analyzed smoking related diseases, such as stress disorders, peripheral vascular diseases, atherosclerosis, and stroke, which could more directly affect heart diseases than smoking. 

- Also, in Korea, substance abuse is very rare that we did not need to take consideration. Also, due to its rarity, we could not reach such information by Korean National Health Insurance Service-National Sample Cohort. 

Reviewer 1 #2: The increased number of dyslipidemic patients in the AA group- whether on treatment, potential implications.

Authors’ response:

- As the reviewer pointed out, whether to be treated dyslipidemia could act as a bias. Thus, we adjusted such confounding factor ‘the presence of dyslipidemia’ when seeking hazard ratios, that we could be free from the possibility of the bias. 

Reviewer 1 #3: The authors may discuss the mechanistics of AA and heart disease, a little more in depth (some authors have hinted at a benefit, as well).

Authors’ response:

- So far, the detailed mechanisms of alopecia areata are unclear. However, we could have been postulated that AA could be related to heart diseases by the various findings (Wang E.H., et al. Alopecia Areata is Associated with Increased Expression of Heart Disease Biomarker Cardiac Troponin I. Acta Derm Venereol. 2018;98(8):776-82; Wang E. et al. Development of autoimmune hair loss disease alopecia areata is associated with cardiac dysfunction in C3H/HeJ mice. PLoS One. 2013;8(4):e62935), and conducted this study. To deliver such perspective more clearly, we added this concept in the INTRODUCTION (Line 56-58)

Reviewer 2 #1: Unclear as to how ethnic differences would affect any potential association between AA and HD. The authors briefly mention this but do not elaborate.

Authors’ response:

- As we can see in other studies (Lane et al, Ethinic differences in blood pressure and the prevalence of hypertension in England. Journal of Human Hypertension 2002; 16: 267–273; Rampal et al, Ethinic differences in the prevalence of metabolic syndrome: Results from a multi-ethinic population-based survey in Malaysia, Plos One, 2012; 7(9): e46365), prevalence of various metabolic disease differs by the ethnicity. Because of the cultural and genetic differences, vulnerability to such diseases which are largely related to heart diseases could be varied by ethnicity. 

- Thus, we were about to confirm the irrelevance between AA and cardiovascular disease, further general heart diseases by the large population of the Korean. 

- To deliver our intention clearly, we inserted such contents in the INTRODUCTION. (line 61-62)

Reviewer 2 #2: Arbitrary definition of long-standing AA as greater than one year.

Authors’ response:

- As approximately 60% of AA patients have at least a partial hair regrowth by 1 year (according to Kang et al. Fitzpatrick’s Dermatology, 9th edi.), previous studies (McDonald Hull et al. Guidelines for the management of alopecia areata. British J Dermatol 2003;149: 692–699; Pratt et al. Alopecia areata. Nat Rev Dis Primers. 2017; 3:17011) have used such criteria for ‘long-standing AA’. Thus, we defined ‘long-standing AA’ as more than a year, as well. 

Reviewer 2 #3: Authors do not clarify why they chose a 1:5 control ratio and if a smaller (or larger) ratio would have had an impact on the findings.

Authors’ response:

- According to Hennessy et al. statistical power can be achieved when more than five controls per case were analyzed in case-control study design. (Hennessy et al. Factors Influencing the Optimal Control-to-Case Ratio in Matched Case-Control Studies. American Journal of Epid 1999; 149(2))

Reviewer 2 #4: Dyslipidemia was more common in AA group but it would have biased the results in favor of an association. Hence, this is likely not relevant.

Authors’ response:

- As the reviewer pointed out, whether to be treated dyslipidemia could act as a bias. Thus, we adjusted such confounding factor ‘the presence of dyslipidemia’ when seeking hazard ratios, that we could be free from the possibility of the bias. 

Reviewer 2 #5: ICD code used for heart failure needs to be identified.

Authors’ response:

- Following this comment, we elaborated the ICD codes for heart diseases in METHODS (line 86-88).

Reviewer 2 #6: Chronic myocardial infarction is a poorly defined entity. I suspect the authors were referencing ICD I25.2 (old myocardial infarction).

Authors’ response:

- According to ICD-10 version of 2019, I25 code (chronic ischaemic heart disease) consists of I25.0 (Atherosclerotic cardiovascular disease, so described), I25.1 (Atherosclerotic heart disease), I25.2 (Old myocardial infarction), I25.3 (Aneurysm of heart), I25.4 (Coronary artery aneurysm and dissection), I25.5 (Ischaemic cardiomyopathy), I25.6 (Silent myocardial ischaemia), I25.8 (Other forms of chronic ischaemic heart disease), and I25.9 (Chronic ischaemic heart disease, unspecified)

- As those disease entities could be both a cause and effect of chronic myocardial infarction, we wanted to include all the subclassification codes in this study. To clarify this intention, we added the ICD-10 code description in METHODS. (line 86-88)

---

## [Decision Letter · Decision Letter 1]

5 Apr 2021

Alopecia areata is not a risk factor for heart diseases: A 10-year retrospective cohort study

PONE-D-21-00164R1

Dear Dr. Choi,

We’re pleased to inform you that your manuscript has been judged scientifically suitable for publication and will be formally accepted for publication once it meets all outstanding technical requirements.

Kind regards,

Feroze Kaliyadan, M.D.

Academic Editor

PLOS ONE

---

## [Editor Report · Acceptance letter]

27 Apr 2021

PONE-D-21-00164R1 

Alopecia areata is not a risk factor for heart diseases: A 10-year retrospective cohort study 

Dear Dr. Choi:

I'm pleased to inform you that your manuscript has been deemed suitable for publication in PLOS ONE. Congratulations! Your manuscript is now with our production department. 

Kind regards, 

on behalf of

Dr. Feroze Kaliyadan 

Academic Editor

PLOS ONE